# The Influence of the Number of Spiking Neurons on Synaptic Plasticity

**DOI:** 10.3390/biomimetics8010028

**Published:** 2023-01-11

**Authors:** George-Iulian Uleru, Mircea Hulea, Alexandru Barleanu

**Affiliations:** Department of Computer Engineering, Gheorghe Asachi Technical University of Iași, Dimitrie Mangeron 27, 700050 Iași, Romania

**Keywords:** spiking neural networks, Hebbian learning efficiency, post-tetanic potentiation, long-term potentiation

## Abstract

The main advantages of spiking neural networks are the high biological plausibility and their fast response due to spiking behaviour. The response time decreases significantly in the hardware implementation of SNN because the neurons operate in parallel. Compared with the traditional computational neural network, the SNN use a lower number of neurons, which also reduces their cost. Another critical characteristic of SNN is their ability to learn by event association that is determined mainly by postsynaptic mechanisms such as long-term potentiation. However, in some conditions, presynaptic plasticity determined by post-tetanic potentiation occurs due to the fast activation of presynaptic neurons. This violates the Hebbian learning rules that are specific to postsynaptic plasticity. Hebbian learning improves the SNN ability to discriminate the neural paths trained by the temporal association of events, which is the key element of learning in the brain. This paper quantifies the efficiency of Hebbian learning as the ratio between the LTP and PTP effects on the synaptic weights. On the basis of this new idea, this work evaluates for the first time the influence of the number of neurons on the PTP/LTP ratio and consequently on the Hebbian learning efficiency. The evaluation was performed by simulating a neuron model that was successfully tested in control applications. The results show that the firing rate of postsynaptic neurons post depends on the number of presynaptic neurons pre, which increases the effect of LTP on the synaptic potentiation. When post activates at a requested rate, the learning efficiency varies in the opposite direction with the number of pres, reaching its maximum when fewer than two pres are used. In addition, Hebbian learning is more efficient at lower presynaptic firing rates that are divisors of the target frequency of post. This study concluded that, when the electronic neurons additionally model presynaptic plasticity to LTP, the efficiency of Hebbian learning is higher when fewer neurons are used. This result strengthens the observations of our previous research where the SNN with a reduced number of neurons could successfully learn to control the motion of robotic fingers.

## 1. Introduction

Spiking neural networks (SNNs) benefit from biological plausibility, fast response, high reliability, and low power consumption when implemented in hardware. An SNN operates using spikes that are the effect of neuronal activation that occurs if a given threshold is exceeded and provides the SNN with sensitivity to event occurrence [1,2]. Thus, one of the critical advantages of SNN over traditional convolutional neural networks is the introduction of time in information processing. Another characteristic of SNN is their ability to learn, which is related to time based on the relative occurrence of the events.

### 1.1. Long-Term Plasticity

The main mechanism that determines learning is the long-term potentiation (LTP) that strengthens the synapses when the presynaptic neuron (pre) activates before the stimulated postsynaptic neuron (post). The reversed order of post and pre activation reduces the synaptic weights via long-term depression (LTD) [3]. The amplitude of the synaptic change due to pairs pre−post and post−pre depends strongly on the temporal difference between the activation of pre and post, respectively [4]. A detailed study of the biological neurons in vitro showed that LTP and LTD windows are asymmetric, making the LTP to dominate the LTD [5], implying that the resultant effect is LTP. Indeed, the essential mechanism of learning in the hippocampus is LTP, as stated in neuroscience [6]. LTP is also the basic element of Hebbian learning that determines the potentiation of the weak synapses if these are paired with strong synapses that activate postsynaptic neurons [7]. This implies that Hebb’s rules are critical for learning in the human brain and they are the foundation for the most biologically plausible supervised SNN learning algorithm [8,9].

### 1.2. Hebbian Learning in Artificial Systems

Besides the biological importance of Hebbian learning, these rules are used in artificial systems, mainly for training competitive networks [8] and to store memories in Hopfield neural networks [10,11]. Hebb’s rule strengthens neural paths that have temporal correlations between pre and post activation. This implies that each neuron tends to pick out its own cluster of neurons, of which the activation is correlated in time by potentiating the synapses that contribute to the activation of postsynaptic neurons [12,13]. In this case, each neuron competes to respond to a subset of inputs matching the principles of competitive learning [8]. In addition, recent research showed that Hebbian learning is suitable to train SNNs of high biological plausibility to control robotic fingers using external forces mimicking the principles of physical guidance [14]. Here, the effect of the strong synapses that are driven by sensors was associated with the effect of weak synapses driven by a command signal [15].

Supervised learning based on gradient descent is more powerful than Hebbian learning [8] in computational applications. However, these error-correcting learning rules are not suitable for bioinspired SNNs because the explicit adjustment of the synaptic weight is not typically feasible. Therefore, for adaptive systems of high biological plausibility, Hebb’s rules are more suitable to train the synapses unsupervised, as they are trained in the brain.

The repetitive activation of pres is independent of a post activity increase in synaptic efficacy by the presynaptic elements of learning such as post-tetanic potentiation (PTP) [16] that can last from a few seconds to minutes [17]. This synaptic potentiation represents an increase in the quantity of the mediator released from the presynaptic membrane during pre activation [16,18]. PTP influences the motor learning specific to Purkinje cells that plays a fundamental role in motor control [19]. Taking into account that this type of presynaptic long-term plasticity occurs in the absence of postsynaptic activity, the Hebbian learning mechanisms are altered by PTP [18].

### 1.3. The Number of Neurons in SNNs

Each neuron is a complex cell comprising multiple interacting parts and small chambers containing molecules, ions, and proteins. The human brain is composed of neurons in the order of $10^{11}$ connected by about $10^{15}$ synapses. Creating mathematical and computational models would be an efficient solution towards understanding the functions of the brain, but even if with an exponential increase in computational power, it does not seem achievable in the near future. Even if this could be achieved, the complexity of the resulting simulation maybe as complex as the brain itself. Hence, there is a need for tractable methods for reducing the complexity while preserving the functionality of the neural system. The size effect in SNNs has various approaches. Statistical physics formalism based on the many-body problem was used to derive the fluctuation and correlational effects on finite networks of *N* neurons as a perturbation expansion of 1/N around the mean field limit of N→∞ [20]. Another method that was used to optimise the size and resilience of the SNN is with empirical analysis using evolutionary algorithms. Thus, smaller networks may be generated by using a multiobjective fitness function incorporating a penalty for the number of neurons evaluating every network in a population [21].

In addition, research on computational neural networks showed that, for classification problems, SNNs use fewer neurons than the second generation of artificial neural networks (ANNs) does [22]. In addition, the hardware implementation of SNNs demonstrated their efficacy in modelling conditional reflex formation [23] or in controlling the contraction of artificial muscles composed of shape memory alloy (SMA). In later applications, SNNs with only a few excitatory and inhibitory neurons have been able to control the force [24,25] and learn the motion [14,15] of anthropomorphic fingers. In addition, using fewer neurons is important in reducing the cost and increasing the reliability of the hardware implementation of SNNs.

Analysing the Hebbian learning efficiency of adaptive SNN provides a useful tool for reducing the size of experimental networks and minimising the simulation time while preserving bioinspired features.

### 1.4. The Goal and Motivation of the Current Research

The presynaptic long-term plasticity determined by PTP reduces the efficiency of Hebbian learning that is determined by LTP. This mechanism is critical to make the neural network respond to concurrent events by potentiating the untrained synapses when activated with the trained neural paths. Thus, PTP potentiates synapses in the absence of a postsynaptic response, meaning that the causality is broken.

Considering these aspects, the goal of this paper is to determine in which conditions the effect of LTP over PTP is maximised, increasing the efficacy of Hebbian learning. Typically, fewer neurons must fire at a higher rate or have stronger potentiated synapses to activate post above the preset rates. Reducing the number of neurons can increase both the firing rate of pres and the synaptic weights that are necessary to reach the requested frequencies of post.

At certain firing rates and synaptic weights, the ratio between the LTP and PTP rates can be higher, implying that associative learning is more efficient. Considering that $dW_{LTP}$ represents the maximal contribution of LTP and $dW_{PTP}$, the effect of PTP to the synaptic weight during the period $t_{L}$ of training, then there is a maximal ratio: $r_{W}^{MAX}$ of $dW_{LTP}^{rMAX}$ and $dW_{PTP}^{rMAX}$. In this work, we consider that the maximal efficiency of Hebbian learning is for $r_{W}^{MAX}$, which corresponds to $dW_{LTP}^{rMAX}$. If a target frequency $f_{POST}$ for the postsynaptic neuron is requested, then a minimal number of untrained presynaptic neurons $n_{UNT}$ with weight $dW_{LTP}^{rMAX}$ can be activated to reach $f_{POST}$. Therefore, the ideal case when LTP is maximal $n_{UNT}$ depends on the functions that describe the weight variation by PTP and LTP, and on $f_{POST}$.

Starting from these ideas, the contribution of this work is twofold: (i) The quantification of Hebbian learning efficiency as the ratio between LTP and PTP; (ii) the evaluation of the influence of the number of neurons on the efficiency of Hebbian learning, focusing on an SNN with reduced number of neurons (fewer than 20 per area).

As presented in Section 1.2 and Section 1.3, there are several comprehensive studies related to Hebbian learning or focused on the influence of variation in the number of neurons on the performance of adaptive SNN. However, there are no studies focused on overlapping these two research directions in systems of high biological plausibility.

The rest of the paper is organised as follows: Section 2 presents the general structure of the neural network and the experimental phases focusing on the proposed neuron model, and on the implementation of PTP and LTP mechanisms. The experimental results along with the details for each measured item are presented in Section 3. The paper ends with Section 4, which discusses the results, focusing on the biological plausibility of the used model, and presents some considerations for future research.

## 2. Materials and Methods

The SNN is based on a neuronal model of high biological plausibility [14]. Although this electronic neuron was implemented and tested in PCB hardware, the analysis presented in this work was based on spice simulations of the electronic circuit.

### 2.1. The Model of the Artificial Neuron

An artificial neuron includes a SOMA and one or more synapses. The electronic SOMA models elements related to information processing, such as the temporal integration of incoming stimuli, the detection of the activation threshold, and a refractory period. Electronic synapses model presynaptic elements of learning such as PTP and the postsynaptic plasticity that determines Hebbian learning via long-term potentiation (LTP). In addition, a synapse stores the synaptic weight using a capacitor that can be charged or discharged in real time using cheap circuits [26,27]. Electronic synapses generate excitatory or inhibitory spikes to feed the corresponding posts.

Figure 1 shows the main elements related to learning that are included by the neuronal schematic that is detailed in Appendix A [14]. The neuron detects the activation threshold using the transistor $T_{M}$, and, during activation, $T_{S}$ generates a spike of variable energy $E_{SPK}$ that depends on the synaptic weight $w_{s}$. In this work, we refer to $w_{s}$ as the voltage $V_{W}$ read in the capacitor $C_{L}$, shown in Figure 1. The synapse is potentiated by PTP that is modelled by the discharge of $C_{L}$ when the neuron activates, and by LTP, which alters the charge in capacitor during the activation of post.

Potential $V_{W}$ determines the duration of the generated spike at OUT, modelling the effect of synaptic weight on postsynaptic activity. Spike duration $t_{SPK}$ is determined by potential $V_{W}$ because, during SOMA activation, transistor $T_{S}$ is open as long as $V_{U}$ (which is proportional with $V_{W}$) is below the emitter-base voltage of $T_{S}$. The variation in $V_{U}$ is given by:(1)$$V_{U}=V_{U_{0}}+\left(V_{DD}-V_{U_{0}}\right)e^{\frac{-t}{R_{D}C_{U}}}$$

The initial potential $V_{U_{0}}$ is calculated using Equation (Equation 2) for cut-off and Equation (Equation 3) for saturated regimes of transistor $T_{S}$ as follows:(2)$$V_{U_{0}}=\frac{V_{W}R_{U}}{R_{U}+R_{D}}$$
(3)$$V_{U_{0}}=\frac{\frac{V_{W}}{R_{U}}+\frac{V_{B}}{R_{B}}}{\frac{1}{R_{D}}+\frac{1}{R_{U}}+\frac{1}{R_{B}}}$$

In $V_{W}=V_{DD}+V_{A}-V_{F}$ and $V_{B}=V_{E}-V_{EB}$, $V_{F}$ and $V_{EB}$ are the forward and emitter base voltages, respectively. Similarly, after SOMA inactivation, $V_{U}$ is restored to $V_{DD}$ as follows:(4)$$V_{U}=\left(V_{DD}-V_{U_{I}}\right)\left(1-e^{\frac{-t}{\left(R_{C}+R_{D}\right)C_{U}}}\right)$$
where $V_{U_{I}}$ is the initial value of $V_{U}$ when SOMA inactivates and $C_{U}$ starts charging.

### 2.2. Model for PTP and LTP

The activation of the neuron that lasts $t_{SPK}=44us$ reverses the polarity of $C_{L}$, which is discharged by an amount that is given by:(5)$$\Delta V_{PTP}=V_{DD}e^{\frac{-t_{SPK}}{\left(R_{D}+R_{U}\right)C_{L}}}$$

Equation (Equation 5) models the potentiation by PTP of the synapse. For expressing LTP, we should consider the charge variation in $C_{L}$ when it is discharged using $C_{A}$, followed by a reset of the charge in $C_{A}$ during post activation.

During neuronal activation, the potential in the capacitor $C_{L}$ varies as follows:(6)$$\Delta V=V_{DD}\left(1-e^{\frac{-t_{SPK}}{R_{A}C_{E}}}\right)$$
where the equivalent capacitance is:$$C_{E}=\frac{C_{L}C_{A}}{C_{L}+C_{A}}$$

Considering that $\Delta V_{A}$ and $\Delta V_{C}$ represent the variation in potential in $C_{A}$ and $C_{L}$, respectively, we denote the ratio:$$k=\frac{\Delta V_{A}}{\Delta V_{C}}=\frac{C_{A}}{C_{L}}$$

Thus, variation in the potential in $C_{L}$ that represents the temporary potentiation of the synapse is:(7)$$\Delta V_{C}=\frac{C_{E}}{C_{L}}\Delta V$$

During the neuronal idle state after its activation, the resultant variation of the potential in $C_{L}$ and $C_{A}$ is:(8)$$\Delta V_{S}=\Delta Ve^{\frac{-t_{W}}{\left(R_{A}+R_{M}\right)C_{E}}}$$
where $t_{W}$ is the time window $t_{p}ost-t_{p}re$ between the moments of the neuronal activation. $C_{A}$ discharges in $C_{L}$ until the potentials in both capacitors reach equilibrium. This variation restores the synaptic weight to the value that was before activation of the presynaptic neuron. If the postsynaptic neuron fires during the restoration of the synaptic weight, capacitor $C_{A}$ is discharged at a significantly higher rate until equilibrium is reached. Taking into account that $R_{L}<<R_{A}+R_{M}$, the potential variation in $C_{L}$ during post activation is negligible. This implies that the variation in the potential in $C_{L}$ that models weight variation by LTP is:(9)$$\Delta V_{LTP}=\frac{C_{E}}{C_{L}}\Delta V_{S}$$

$\Delta V_{S}$ decreases according to Equation (Equation 8), implying that $\Delta V_{LTP}$ depends on the time window $t_{W}$.

For this neuronal design $w_{s}$ varies in the opposite direction, with $V_{W}$ in the range of [0.2 V:1.6 V]. This implies that a lower $V_{W}$ models higher synaptic potentiation. To simplify the presentation, in this research, we refer to variation dW in the voltage in $C_{L}$ that occurs during potentiation.

Therefore, the experiments presented in the following section focus on evaluating how the number of neurons affect learning efficiency.

### 2.3. The Structure of the SNN

The synaptic configuration includes two presynaptic neural areas, $preNA_{T}$ and $preNA_{UNT}$, which include $n_{T}$ and $n_{UNT}$ neurons, respectively. The pres included in these neural areas connect to only one post, as in Figure 2a. For allowing weight variation by LTP, at the beginning of each experiment, synapses $S_{T}$ between $PreNA_{T}$ and post were fully potentiated for the activation of post, while the weights of synapses $S_{UNT}$ driven by $PreNA_{UNT}$ were minimal. The SNN included additional neurons $Pre_{AUX}$ and $Post_{AUX}$ for the evaluation of the potentiation by PTP of $S_{AUX}$, which had the same value as that for $S_{UNT}$. This allowed for us to compare the PTP and LTP effects in similar conditions.

As shown in Figure 2b, neurons in each presynaptic area were activated by constant potentials $V_{1},\ldots ,V_{N}$ or by pulses, as we detail in the sequel. For modelling the variability in the electronic components, input resistors $R_{1}$, $R_{2}$, and $R_{N}$, shown in Figure 2b, were set in an 10% interval that varied the firing rate of pres slightly.

The firing rate of the postsynaptic neurons $f_{POST}$, and the variation in the synaptic weights $dW_{LTP}$ and $dW_{PTP}$ due to LTP and PTP, respectively, were determined via measurements on the simulated electronic signals [28,29]. The values of the input voltages for the activation of pres were set to several values to activate the neurons in the range that was used during our previous experiments [14]. In order to highlight the influence of the number of neurons on the Hebbian learning efficiency, the initial synaptic weights were minimal for extending their variation range.

### 2.4. Experimental Phases

The experiments started with a preliminary phase in which we determined $dW_{LTP}$ and $dW_{PTP}$ for a single spike when $f_{UNT}$ took several values, and variation in $f_{POST}$ with the number $n_{UNT}$. Following these preliminary measurements, we evaluated the efficiency of Hebbian learning by calculating the ratio $r_{W}=dW_{LTP}/dW_{PTP}$ during several phases as follows:

Phase 1. The value of $dW_{LTP}$ was determined when $n_{T}$ and $n_{UNT}$ in the neural areas $preNA_{T}$ and $preNA_{UNT}$, respectively, varied independently or simultaneously. These results were compared with the effect of PTP when only the neurons in the untrained area $preNA_{UNT}$ were activated. Variation $dW_{LTP}$ in the synaptic weight included potentiation due to LTP being determined by pre−post pair activation and due to PTP that occurred due to pre action potential.

Typically, the frequency of post can be controlled in certain limits by adjusting the firing rate of pres independent of the number of neurons. In order to simplify the SNN structure during Phases 2 and 3, $preNA_{T}$ included one neuron.

Phase 2. Next, we determined the variation in $dW_{LTP}$ and $dW_{PTP}$ when synapses in $preNA_{UNT}$ were trained until they were able to activate post in the absence of $preNA_{T}$ activity.

Phase 3. For the last phase of the experiments, we considered a fixed frequency $f_{M}$ of the output neuron that matched the firing rate of the output neurons that actuated the robotic junctions in our previous experiments [14]. Thus, the SNN was trained until $f_{M}$ had reached 100 Hz when stimulated only by $preNA_{UNT}$ independent of $preNA_{T}$. In order to extract the contribution of PTP to the $dW_{LTP}$, neuron $Pre_{AUX}$ was activated at the same rate with $preNA_{UNT}$, and $dW_{PTP}$ was measured.

For the untrained pres, we set different frequencies that were not divisible with the firing rate of post, mimicking a less favourable scenario of neuronal activation. In this case, the time interval between pre and post activation varied randomly, increasing the diversity of the weight gain per the action potential of post. In a favourable scenario, the frequency of post is the divisor of the firing rate of pre, which improves the weight gain via the synchronisation of neuronal activation.

## 3. Results

The obtained results during the experimental phases mentioned above are presented here.

### 3.1. Preliminary Phase

In order to asses the influence of the electronics on synaptic potentiation during a single spike, we determined the weight variation by PTP for several values of $V_{IN}$ when pre activated once. As presented in Figure 3a, PTP decreased as $f_{UNT}$ increased. However, long term, this variation was compensated by the number of spikes per time unit that increased at a higher rate with $f_{UNT}$. A similar evaluation was performed for LTP when post was activated by trained neurons at t=0.002 s after the activation of the untrained pre. In this case, the influence of LTP presented in Figure 3b was extracted from the measured dW by the subtraction of the PTP effect shown in Figure 3a.

Typically, the output frequency of an SNN depends on the number of pres that stimulate post, as shown in Figure 4a. Starting from this observation, we determined $dW_{LTP}$ after 2 s of training for a different number of pres in $PreNA_{UNT}$ and $PreNA_{T}$ when $f_{UNT}=75\;\mathrm{Hz}$ and $f_{T}=100\;\mathrm{Hz}$. Figure 4b–d show that $dW_{LTP}$ depended on the number of pres following different patterns for the trained and untrained neurons. In addition, the learning rate by LTP increased with the number of pres, mainly due to higher values of $f_{POST}$ determined by the activation of more pres.

### 3.2. The Efficiency of Hebbian Learning

Variation in the ratio $r_{W}$ with the voltage $V_{W}$ synaptic weight is presented in Figure 5a. This represents the ideal case when LTP is maximal, which was obtained by the activation of the postsynaptic neuron shortly after the untrained pre. $r_{W}$ was maximal for a specific weight that was far from the limits of the variation interval.

Weight variation dW for different firing rates of pre was determined for both PTP and LTP when the neurons activated for a fixed period of time t=2 s. The data plotted in Figure 5b show that variation in the ratio $r_{W}$ reduced significantly when the frequency of the untrained pres was above 50 Hz. Thus, taking into account that rW was almost stable for a single neuron in $PreNA_{UNT}$, for the next experimental phase, we evaluated the influence of the number of neurons on rW for fixed activation frequency $f_{UNT}=50$ Hz.

In this setup, the SNN was trained until the first activation of post by the synapse that was potentiated by LTP. The PTP level for the synapse $S_{AUX}$ was determined via the activation of the auxiliary neuron $Pre_{AUX}$ (see SNN structure in Figure 2a) at the same frequency as that of the untrained pres. In order to determine if $r_{W}$ had a similar variation for another frequency of the neurons in the $PreNA_{UNT}$, we performed similar measurements for $f_{UNT}=75$ Hz. As presented in Figure 6, the variation in $r_{W}$ showed that the best learning efficiency was obtained for $n_{UNT}=1neuron$ when $f_{UNT}=75$ Hz and for $n_{UNT}=3neurons$ when $f_{UNT}=75$ Hz. The different number of pres indicated that $f_{UNT}$ influenced the optimal number of neurons for the best learning efficiency when the neural paths were trained until the first activation of post by $PreNA_{UNT}$ independent of $PreNA_{T}$.

The next experimental phase evaluated rW and the duration of the training process $t_{L}$ when the firing rate of the output neurons reached $f_{M}=100$ Hz, while the untrained pres in the area $PreNA_{UNT}$ activated at several firing rates in a set that included divisors of $f_{M}$.

The plots in Figure 7a,b show that weight variations $dW_{PTP}$ and $dW_{LTP}$ decreased when the number of untrained pres
$n_{UNT}$ increased. Typically, $dW_{PTP}$ is proportional with $t_{L}$, implying that the SNN learns faster when more neurons activate post. The improved value for $t_{L}$ was determined by the lower weights that were necessary to activate post at the requested firing rate. In order to compare $dW_{PTP}$ and $dW_{LTP}$, we determined the $w_{s}$ that were potentiated by PTP when the neuron $Pre_{AUX}$ was activated at $f_{UNT}$ as the neurons in the area $PreNA_{UNT}$. Figure 8a shows the variation rW for several firing rates $f_{UNT}$ that were not divisors of $f_{M}$. In this case, local maximum $rW_{local}$ was for a single neuron per area ($n_{UNT}=1$). Taking into account that LTP may be more efficient when $f_{UNT}$ is a divisor of $f_{M}$ due to the synchronisation of the pre−post neurons, we evaluated the weight variation for $f_{UNT}=$ 25, 33.3, and 50 Hz. In order to eliminate the variation in PTP with the continuous input voltage of the neurons, the pres were activated with digital pulses with the same amplitude generated at rate $f_{UNT}$. The results presented in Figure 8b show that the maximal $rW_{local}$ was obtained for $n_{UNT}=2$ neurons. The best learning efficiency $rW_{MAX}$ was obtained when $f_{UNT}=25$ Hz, and the untrained presynaptic area included two neurons.

Typically, the weight variation with training duration $t_{L}$ by LTP and PTP varies, following different patterns. The difference between the two functions implies that there is a value for $t_{L}^{max}$ where the ratio between LTP and PTP is maximal. This value corresponds to a synaptic weight $w_{LTP}$ obtained by LTP and consequently to a potential $V_{W}$. Typically, there is a minimal number of pres $n_{UNT}$ firing at a fixed frequency that are able to activate a post when the weight is $w_{LTP}$. In our work, the best $r_{W}=4.28$ for $n_{UNT}=2$ neurons corresponded to the potential $V_{W}=0.7$ V in the weight capacitor.

## 4. Discussion and Conclusions

At the synaptic level, the neural paths in the brain are trained by associative or Hebbian learning that is based on long-term potentiation, which is the postsynaptic element of learning. From a biological point of view, presynaptic long-term plasticity violates Hebbian learning rules that depend on postsynaptic activity. Previous research on SNN showed that the control systems use a reduced number of electronic neurons, while in classification tasks, SNN uses fewer neurons than the traditional CNN does. Starting from these ideas, this work focused on the evaluation of the influence of the number of neurons on the efficiency of Hebbian learning, characterised as the ratio of LTP and PTP effects on the synaptic weights. This ratio increases the effect of LTP and consequently the power of the SNN to discriminate between the neural paths that are trained by associative learning over the paths where only presynaptic plasticity occurs. The simulation results showed that, despite the fact that LTP depends mainly on the frequency of postsynaptic neurons, the number of neurons affect the Hebbian learning efficiency when the posts must reach a predefined frequency. In this case, the best LTP/PTP ratio was obtained when the frequency of the untrained pres was the lowest divisor of the target frequency of post. The efficiency of Hebbian learning reached a maximum for two pres and decreased in the opposite direction with the number of pres. Taking into account that, for a certain number of neurons, the LTP/PTP ratio was better, we could deduce that certain synaptic weights resulted in better Hebbian learning efficiency. Indeed, the position of the maximal $r_{W}$ inside the variation interval (Figure 8b) matched the variation in the $r_{W}$ in the ideal case presented in Figure 5a. This implies that the minimal number of neurons that were necessary to activate post at the requested firing rate was related to the synaptic weight. In conclusion, previous research showed that electronic SNNs with a reduced number of neurons are trained efficiently by Hebbian learning, while the current research strengthen the idea showing that fewer neurons improve associative learning. This could reduce the cost and improve the reliability of the hardware implementation of SNNs.

## Figures and Tables

**Figure 1 biomimetics-08-00028-f001:**
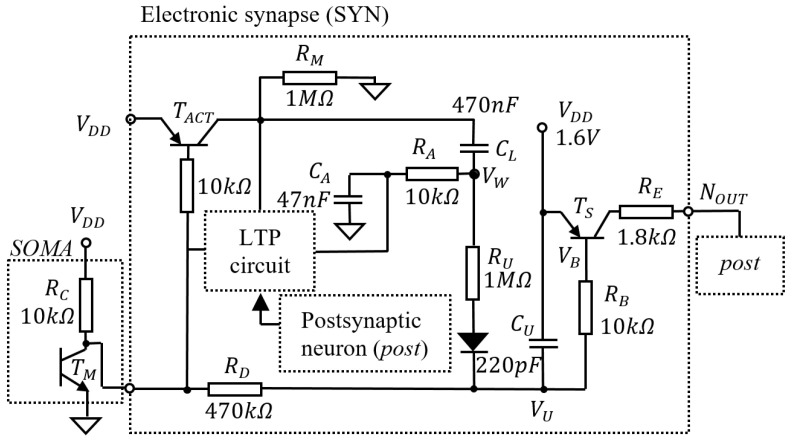
The main elements of the neuronal schematic that are used to model the mechanisms of learning.

**Figure 2 biomimetics-08-00028-f002:**
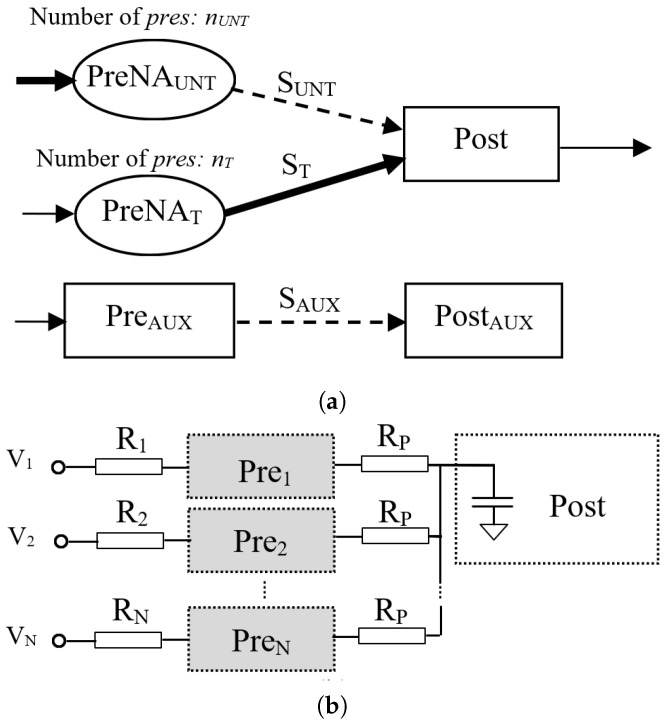
(**a**) The structure of the SNN that was used to perform tests; (**b**) R varied among presynaptic neurons Pre1,Pre2,…,PreN.

**Figure 3 biomimetics-08-00028-f003:**
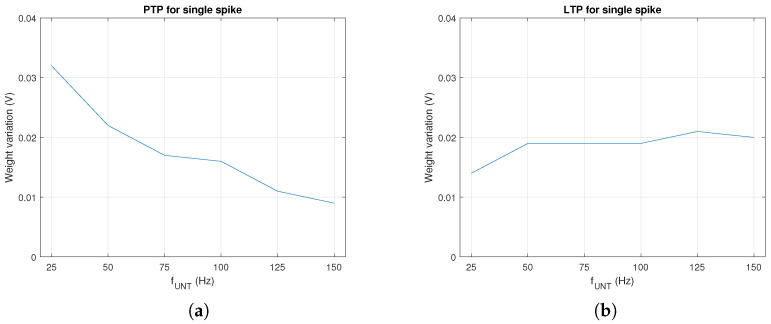
Weight variation with $f_{pre}$ during a single spike by (**a**) PTP and (**b**) LTP for a time window.

**Figure 4 biomimetics-08-00028-f004:**
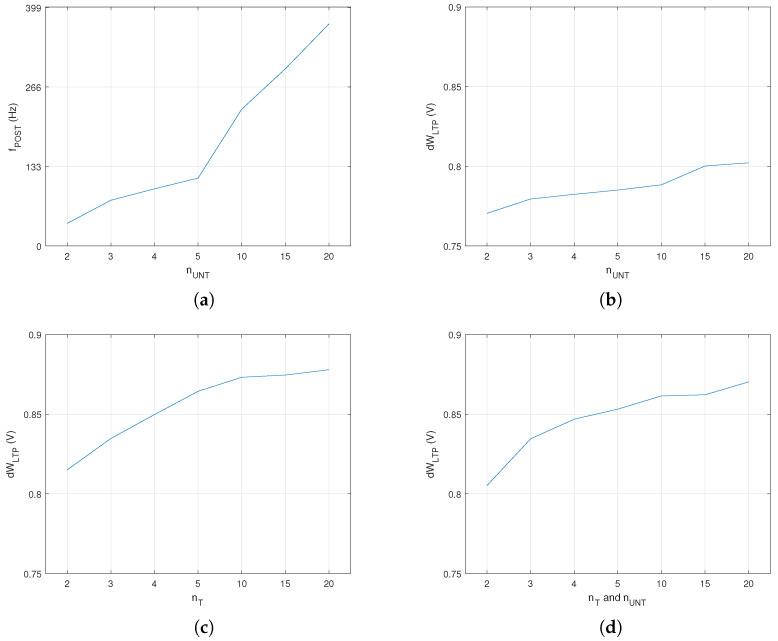
(**a**) The frequency of the output neuron. The synaptic weight variation by LTP after 2 s of training for variation in the number of (**b**) untrained neurons with $f_{UNT}=75$ Hz; (**c**) trained neurons with $f_{T}=100$ Hz; (**d**) both trained and untrained neurons.

**Figure 5 biomimetics-08-00028-f005:**
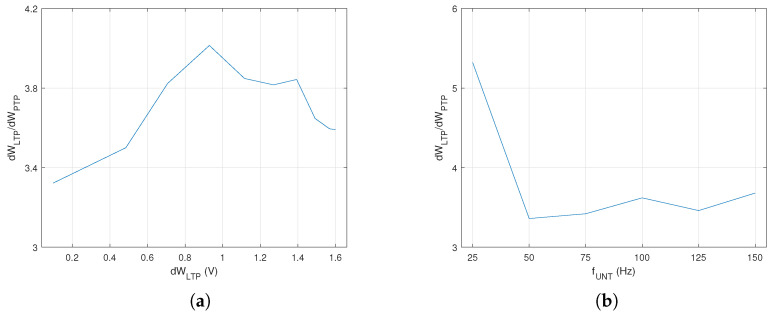
(**a**) Variation in the ratio $r_{W}$ with synaptic weight; (**b**) the ratio of synaptic weight variation between PTP and LTP after 2 s of training.

**Figure 6 biomimetics-08-00028-f006:**
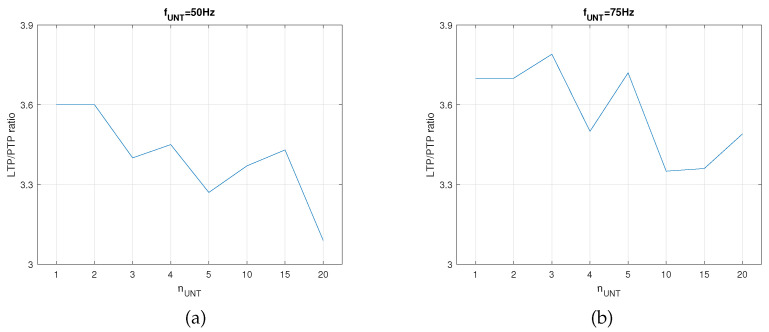
The ratio of potentiation levels between LTP and PTP when the SNN was trained until first activation of post for (**a**) $f_{UNT}=50\;\mathrm{Hz}$ and (**b**) $f_{UNT}=75\;\mathrm{Hz}$.

**Figure 7 biomimetics-08-00028-f007:**
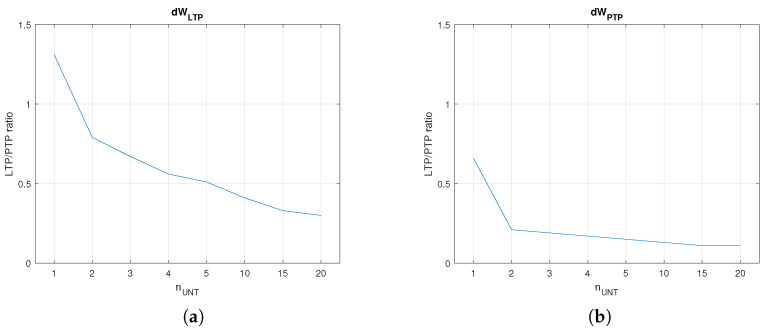
Learning until $f_{POST}=100$ Hz when the number of untrained neurons $n_{UNT}$ varied. (**a**) $dW_{LTP}$ and (**b**) $dW_{PTP}$ for $f_{UNT}=50$ Hz.

**Figure 8 biomimetics-08-00028-f008:**
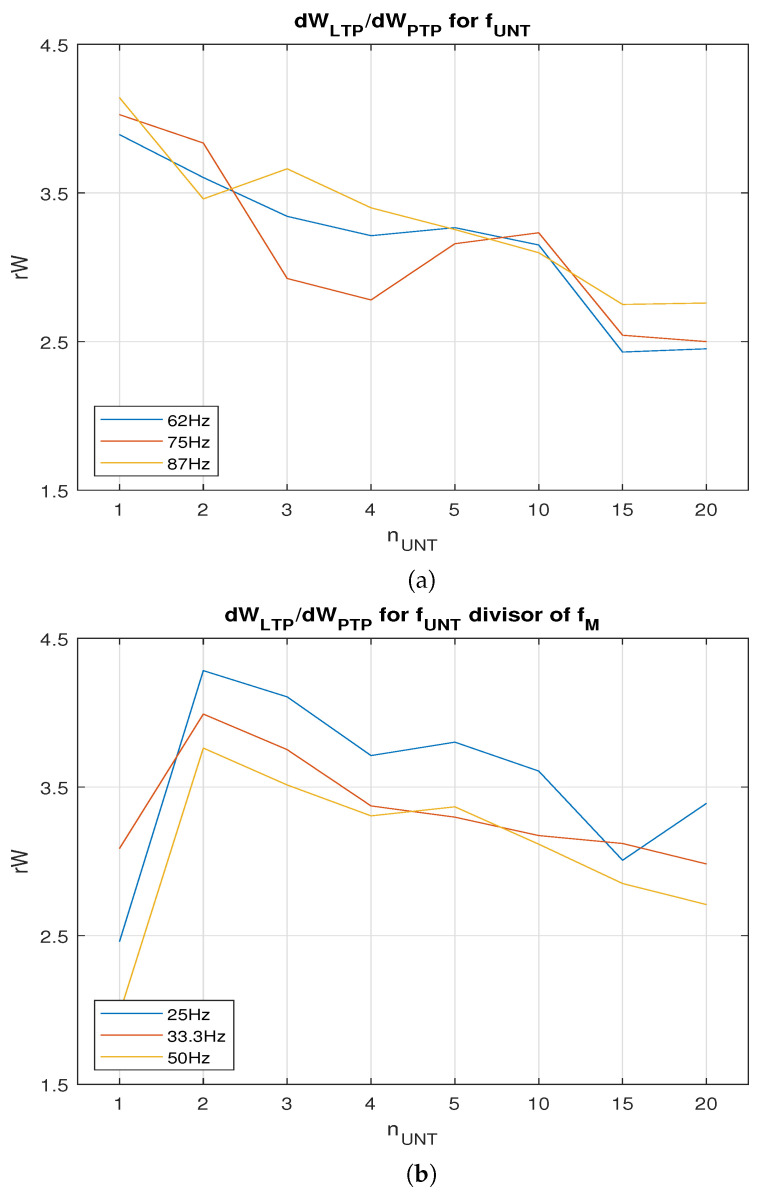
Variation in ratio $rW=dW_{LTP}/dW_{PTP}$ with $n_{UNT}$ when (**a**) $f_{UNT}$ is not a divisor of $f_{M}$ and (**b**) $f_{UNT}$ is a divisor of $f_{M}$.

## Data Availability

Not applicable.

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
