# Peer review of "The Influence of the Number of Spiking Neurons on Synaptic Plasticity"

_biomimetics, 2023, doi:10.3390/biomimetics8010028_

Round 1
Reviewer 1 Report
The main contribution of this paper is using simulations on an electronic neuron model which has been already validated in other works. A real time experimenter can make changes easily and quickly within microseconds without additional cost over increasing hardware processing power or optimizing software codes. Therefore, it was possible to simulate directly real experiments in order to evaluate easily both quantitative results and qualitative aspects such as electrical properties and behavior associated with the proposed approach in addition to providing a new methodology for understanding and developing spiking neural networks.
In order to improve this paper, the authors could add new experiments and illustrate changes related to existing research in order to evaluate these findings further and propose an algorithm based on them:
The algorithm proposed requires a preliminary set up phase where all system parameters (number/firing rates etc.) have been adjusted beforehand according to experimental data obtained from a previous step. It also assumes pre-set values for input values. Additionally, please clarify if it considers finding initial weights between layers with reasonable values and modifications of these weights using a predetermined algorithm (Hebbian) or analyzes each sample with standard statistical tests according to data distribution characteristics (eccentric training) or takes into account best results from previous steps if any similar situations occur during testing time (winner takes all). Other algorithms determine whether or not current input samples would result into increase or decrease for each layer if current weights were modified after testing phase (dynamic adjustment). This should be compared as well. Other approaches run forward simulations to find what kind/rate/pattern would be required for final net-input values considering previously estimated initial inputs along with final inputs given after dynamic adjustment; implement winning/losing scenarios depending on previous output result and corresponding adjustments. Please describe how much did results differ from expected inputs during testing time. How well does our approach follow test results comparing it with previous ones along with total sum error calculated between expected inputs & actual outputs? Compares current test result with new results given after dynamic adjustment phase and gives us opportunity either confirm this action or improve it using additional steps based on such results e g if we get desired result after adjusting it accordingly then we may skip remaining dynamic phases but otherwise we proceed to the next stage where next sample input may be considered instead e g if we got expected result after adjusting but test conditions weren't suitable.
Author Response
We would like to thank the reviewers for their very constructive comments. They are very helpful for revising and improving our paper and allowed us to improve the technical contents and presentation quality. We appreciate the reviewers’ and editor’s efforts very much. We have taken into consideration and complied with all the comments and suggestions. We have studied the comments carefully and provided detailed explanations for our changes in this response.

Reviewer 2 Report
The manuscript sounds technically poor, I have following concerns should be addressed before any decision.
*The existing literature should be classified and systematically reviewed, instead of being independently introduced one-by-one.
*The abstract is too general and not prepared objectively. It should briefly highlight the paper's novelty as what is the main problem, how has it been resolved and where the novelty lies?
*The 'conclusions' are a key component of the paper. It should complement the 'abstract' and normally used by experts to value the paper's engineering content. In general, it should sum up the most important outcomes of the paper. It should simply provide critical facts and figures achieved in this paper for supporting the claims.
*For better readability, the authors may expand the abbreviations at every first occurrence.
*The author should provide only relevant information related to this paper and reserve more space for the proposed framework.
*However, the author should compare the proposed algorithm with other recent works or provide a discussion. Otherwise, it's hard for the reader to identify the novelty and contribution of this work.
*The descriptions given in this proposed scheme are not sufficient that this manuscript only adopted a variety of existing methods to complete the experiment where there are no strong hypothesis and methodical theoretical arguments. Therefore, the reviewer considers that this paper needs more works.
*Key contribution and novelty has not been detailed in manuscript. Please include it in the introduction section
*What are the limitations of the related works
*Are there any limitations of this carried out study?
*How to select and optimize the user-defined parameters in the proposed model?
*There are quite a few abbreviations are used in the manuscript. It is suggested to use a table to host all the frequently used abbreviations with their descriptions to improve the readability
*Explain the evaluation metrics and justify why those evaluation metrics are used?
*Some sentences are too long to follow; it is suggested that to break them down into short but meaningful ones to make the manuscript readable.
*The title is pretty deceptive and does not address the problem completely.
*Every time a method/formula is used for something, it needs to be justified by either (a) prior work showing the superiority of this method, or (b) by your experiments showing its advantage over prior work methods - comparison is needed, or (c) formal proof of optimality. Please consider more prior works.
*The data is not described. Proper data description should contain the number of data items, number of parameters, distribution analysis of parameters, and of the target parameter itself for classification.
* The related works section is very short and no benefits from it. I suggest increasing the number of studies and add a new discussion there to show the advantage
*Use Anova test to record the significant difference between performance of the proposed and existing methods.
Author Response
We would like to thank the reviewers for their constructive comments. They are very helpful for revising and improving our paper and allowed us to improve the technical contents and presentation quality. We appreciate the reviewers’ and editor’s efforts very much. We have taken into consideration and complied with all the comments and suggestions. We have studied the comments carefully and provided detailed explanations for our changes in this response.

Reviewer 3 Report
The paper titled “The influence of the number of spiking neurons on the efficiency of Hebbian learning” evaluate for the first time the influence of the number of neurons on the efficiency of Hebbian learning in an electronic SNN.
I would like providing few major and minor feedback.
Major:
• Despite the authors doing several workbench simulations, it is difficult to follow simulations and outcomes since there is no mathematical model.
• Furthermore, reproducibility could be a better term to characterize the range of values utilized to test the electronics board than the authors' choice of leaving resistors and capacitors in figure 1 without labels is also not good.
• I would recommend providing a brief introduction to hebbian learning and the mathematical model connected to the simulation carried out in this article.
• I would recommend a higher-quality figure (e.x fontsize, etc)
• The reader initially believed that this study was a silico simulation, however the conclusion clarifies that the experiment was carried out on a PCB board. In the end, setup, equipment, experiment implementation, etc., may be explained.
• I would advise refocusing the Introduction section. The introduction is weak, and many of the themes aren't truly related. The reader also finds it difficult to draw a connection between the work done in this study and machine learning.
• Additionally, the reader may not truly understand how the topic of this paper may be connected to "SMA actuators that drive...". Additionally, the reader of this paper could be unfamiliar with SMA, thus what SMA is ?
• "Movement coordination" sounds unexpected in line 51. I'd advise addressing.
• A section describing setup, parameter setting, data collecting, analysis software, etc. should be added, in my opinion.
Minor
• I would propose focusing more on the meaning of long sentences, typos, and acronyms.
• I would suggest mathematical models and high-resolution figures.
• I would recommend enhancing sentence's language.
Author Response

(The authors gave the same response as above.)

Round 2
Reviewer 2 Report
The paper is relatively improved. However, still needs some improvements.
1The descriptions given in this proposed scheme are not sufficient that this manuscript only adopted a variety of existing methods to complete the experiment where there are no strong hypothesis and methodical theoretical arguments. Therefore, the reviewer considers that this paper needs more works.
2The related works section is very short and no benefits from it. I suggest increasing the number of studies and add a new discussion there to show the advantage.
3The manuscript is not well organized. The introduction section must introduce the status and motivation of this work and summarize with a paragraph about this paper.
Author Response
Dear distinguished Reviewer,
We would like to thank you very much for your effort and for the very constructive comments. They are very helpful for revising and improving our paper and allowed us to improve the technical contents and presentation quality.We have taken into consideration and complied with all the comments and suggestions. We have studied the comments carefully and provided detailed explanations for our changes in this response.
Respectfully yours,
George Iulian Uleru
Mircea Hulea
Alexandru Barleanu
